# A Mixture-of-Expert Approach to RL-based Dialogue Management

**Yinlam Chow**          **Aza Tulepbergenov**          **Ofir Nachum**          **Dhawal Gupta**

**Moonkyung Ryu**          **Mohammad Ghavamzadeh**          **Craig Boutilier**

Google Research
`{yinlamchow, atulep, ofirnachum, dhawgupta, mkryu, ghavamza, cboutilier}@google.com`

## Abstract

Despite recent advancements in language models (LMs), their application to dialogue management (DM) problems and ability to carry on rich conversations remain a challenge. We use reinforcement learning (RL) to develop a dialogue agent that avoids being short-sighted (outputting generic utterances) and maximizes overall user satisfaction. Most existing RL approaches to DM train the agent at the word-level, and thus, have to deal with a combinatorially complex action space even for a medium-size vocabulary. As a result, they struggle to produce a successful and engaging dialogue even if they are warm-started with a pre-trained LM. To address this issue, we develop a RL-based DM using a novel *mixture of expert* language model (MoE-LM) that consists of (i) a LM capable of learning diverse semantics for conversation histories, (ii) a number of *specialized* LMs (or *experts*) capable of generating utterances corresponding to a particular attribute or personality, and (iii) a RL-based DM that performs dialogue planning with the utterances generated by the experts. Our MoE approach provides greater flexibility to generate sensible utterances with different intents and allows RL to focus on conversational-level DM. We compare it with SOTA baselines on open-domain dialogues and demonstrate its effectiveness both in terms of the diversity and sensibility of the generated utterances and the overall DM performance.

## 1 Introduction

With the tremendous advancements in natural language understanding and generation, increasing attention has been directed to construct intelligent dialogue agents that can carry out engaging conversations with users. Such interactions can be open-ended, contain different topics, and often involve an underlying task, such as negotiation, information exchange, and recommendation. Therefore, to satisfy the user, a good dialogue agent should not only generate natural responses, but also be capable of pursuing the task's objectives and adapting to the user's feedback on-the-fly.

A standard solution is to train the dialogue agent using behavioral cloning, where the agent is a language model (LM) that imitates the utterances in the training set (Gašić et al., 2011; Fatemi et al., 2016). By leveraging deep neural networks, e.g., RNNs (Sutskever et al., 2014) and Transformers (Vaswani et al., 2017), a LM encodes the conversation to a low-dimensional dialogue state and predicts an utterance, but steering such generation for particular purposes remains an open question. Several works studied ways to fine-tune a LM to generate texts with specific contexts (Ziegler et al., 2019; Ficler and Goldberg, 2017). Other results learned a single steerable LM that is capable of generating utterances for multiple specific intents (Gu et al., 2017; Chen et al., 2018; Subramani et al., 2019; Dathathri et al., 2019). While these LMs produce fluent and relevant responses, it is unclear how to control them to systematically pursue goals during multi-turn dialogue conversations.

Another popular approach is to view dialogue management (DM) as a control problem and use reinforcement learning (RL) to optimize the agent's policy (which is often a LM itself). Using RL for dialogue systems has a long history. Earlier work relies on specific, hand-crafted semantic states (Levin and Pieraccini, 1997; Singh et al., 2002; Walker, 2000) or partially observable belief states (Williams and Young, 2007; Young et al., 2010), in which the agent chooses the best hand-crafted dialogue act at each turn, with the goal of either satisfying the user (Shah et al., 2018),

completing the task (Shi and Yu, 2018), or responding to the user's query (Serban et al., 2017a). However, the application of these approaches is limited to problems whose action space can be captured by hand-crafted representations, and they cannot handle complex conversations. On the other hand, more recent approaches use deep learning to extract semantic representations from conversation histories, treat these representations as dialogue belief states, and apply RL to learn a word-level generative DM agent (Jaques et al., 2019; Li et al., 2016; 2017; Shin et al., 2020). However, since there are innumerable possibilities of language utterances, and thus, the action space of the RL problem is extremely large, the agent often performs planning poorly and generates incomprehensible utterances (Zhao et al., 2019). Another issue is that RL only optimizes a scalar reward, while the aforementioned methods often need to optimize for both the quality of the generated utterance, e.g., ease of answering (Li et al., 2016), fluency (Li et al., 2017; 2019), and diversity (Yarats and Lewis, 2018), and the goal, e.g., conversation length (Zhou et al., 2020), user's sentiment (Hancock et al., 2019), and task completion (Verma et al., 2022; Jang et al., 2021). Moreover, defining the reward as weighted combination of these metrics is not ideal, since the hand-picked weights do not often reflect the underlying success criteria.

To address the above issues related to using RL in dialogue management (DM) systems, we propose an RL-based DM agent using a novel *mixture of expert* (MoE) approach. Our MoE approach is based on a mixture of expert language model (MoE-LM), which consists of three main components: **1)** a LM (a probabilistic encoder and a decoder) capable of learning diverse semantics for conversation histories, and as a result generating diverse utterances, which we refer to as the *primitive* LM or $\text{LM}_0$, **2)** a number of *specialized* LMs (or *experts*), $\{\text{LM}_i\}_{i=1}^m$, that each is constructed using the latent space learned by $\text{LM}_0$, but has been trained such that it is capable of generating utterances corresponding to a certain intent or personality, and **3)** an RL-based dialogue manager (DM) that at each turn, given the latent state shared by the experts $\{\text{LM}_i\}_{i=0}^m$ and the utterance action(s) they suggest, chooses one among them for the agent to execute. Our MoE-LM can be seen as a special case of hierarchical LMs (e.g., Serban et al. 2017a; Zhao et al. 2019; Saleh et al. 2020), but it is different than them because it learns both the LMs (experts) and the DM. Moreover, the DM in MoE-LM is a policy conditioned on both the latent state and the actions suggested by the experts, and not just the state as it is common in hierarchical RL. The primitive LM ($\text{LM}_0$) plays an important role in this model because it learns diverse semantics for conversation histories and allows the agent to generate a wide variety of utterances. This diversity is also shared with the specialized LMs (experts) and gives them flexibility in generating their (more) specialized utterances. Another important feature of MoE-LM is its modularity that facilitates adding and removing specialized LMs (experts). Moreover, this hierarchical architecture allows us to solve an RL problem with much smaller state and action spaces, which is quite important in the quality of the learned policy. Finally, since the candidate utterances are generated by experts with different intents, instead of combining all agent-user signals into a single RL reward, our DM agent can focus on optimizing the specific goal of the conversation task.

We start the paper with a brief introduction of LMs and the use of Markov decision processes (MDPs) in modeling dialogue management problems in Section 2. We then describe the overall architecture of our MoE-LM in Section 3, followed by the detailed implementation of each of its three main components (described in the above paragraph) in Sections 4 to 6. Finally, in Section 7, we demonstrate the effectiveness of our MoE-LM in open-domain dialogues, in terms of both its ability to generate diverse and sensible utterances and its overall DM performance.

## 2 PRELIMINARIES

**Language Models (LMs)** In this work, we employ seq2seq LMs to generate the next utterances in a dialogue. We assume access to a dataset of the form $\mathcal{D} = \{(\mathbf{X}^{(k)}, Y^{(k)})\}_{k=1}^{|\mathcal{D}|}$, where each $\mathbf{X} = \mathbf{X}^{(k)}$ is a $L$-turn conversation history $\mathbf{X} = \{X_l\}_{l=0}^{L-1}$ and $Y$ is its next utterance. We denote by $N_{\mathbf{X}}$, an upper-bound on the length (number of tokens) of each utterance $X_l$ in $\mathbf{X}$.[1] The role of a LM is to predict the probability of the next utterance $Y$, consisting of $N$ tokens, conditioned on the conversation history $\mathbf{X}$, i.e., $p(Y = \{y_n\}_{n=1}^N \mid \mathbf{X})$. In the transformer architecture (**?**), the LM first encodes the conversation history $\mathbf{X}$ using an encoder $\Phi$ to a $(L \times N_{\mathbf{X}})$-length sequence of embeddings $\{(z_{l,0}, \ldots, z_{l,N_{\mathbf{X}}-1})\}_{l=0}^{L-1}$, where each $z_{l,n}$ is a vector in the latent space. For notational convenience, we concatenate these embeddings into a single embedding $z \in \mathcal{Z} \subseteq \mathbb{R}^d$ and denote the overall dimension of the latent space as $d$. In the RNN architecture (Serban et al., 2016), the LM's encoder $\Phi$ directly maps the conversation history $\mathbf{X}$ to a latent state $z \in \mathcal{Z} \subseteq \mathbb{R}^d$. In both architectures, the next utterance $\widehat{Y} = \{\widehat{y}_n\}_{n=1}^N$ is sampled token-by-token from the decoder $\Psi$,

---

[1]If the actual utterance $X_l$ has fewer tokens than $N_{\mathbf{X}}$, it will be padded by a specific token and masked.

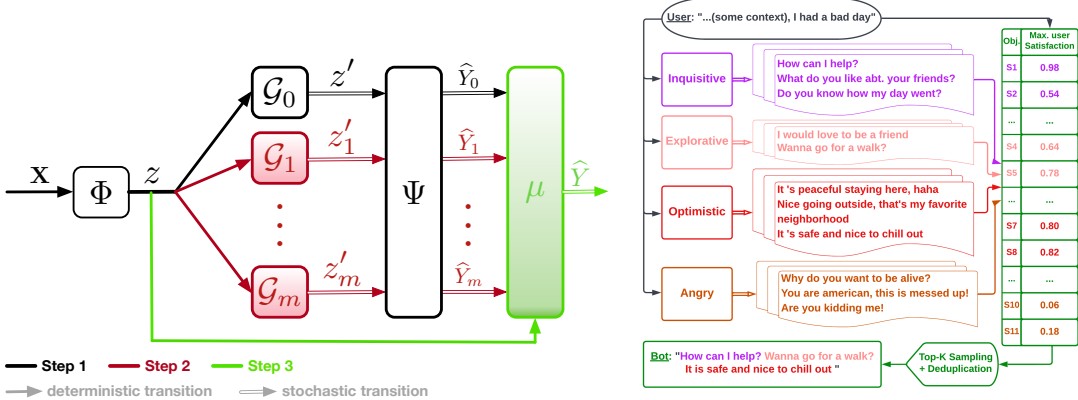

Figure 1: (Left) MoE-LM Architecture. (Right) Sample utterance workflow generated by an MoE-LM trained with Reddit data. Step 1: $\Phi$ encodes conversation history. Step 2: $\Psi \circ \mathcal{G}_i, \forall i$, generate candidate bot utterances. Step 3: $\mu$ selects the bot response by $Q$-score ranking & post-processing.

i.e., $\widehat{Y} \sim \Psi(\cdot \mid z) = \prod_{n=1}^{N} \Psi(\widehat{y}_n \mid \widehat{y}_0, \dots, \widehat{y}_{n-1}; z)$, where $\widehat{y}_0$ is a fixed initial (start-of-sentence) token (Chien and Kuo, 2019), and the latent state is denoted as $z = \Phi(\mathbf{X})$.[2]

**Markov Decision Processes (MDPs)** have been used to model dialogue management problems in a variety of settings (Li et al., 2016; Asadi and Williams, 2016; Jaques et al., 2019). In such MDPs, denoted by $\mathcal{M} = (\mathcal{S}, \mathcal{A}, P, r, s_0, \gamma)$, the state space $\mathcal{S}$ represents the tokenized conversation history and the initial state $s_0 \in \mathcal{S}$ is the initial user's query. The action space $\mathcal{A}$ is also the tokenized language space with each action $a \in \mathcal{A}$ being the agent's next utterance (which is a fixed-length, $N_\mathbf{X}$, sequence of tokens). The transition kernel $P$ models the user's response to the action taken by the agent (bot). Finally, the reward function $r$ measures the user's satisfaction. In these MDPs, we can think of the entire LM as a policy that maps conversation histories to next utterances, and solve them by finding a policy $\pi^*$ with maximum expected discounted return, i.e., $\pi^* \in \arg\max_\pi J_\pi := \mathbb{E}[\sum_{k=0}^{\infty} \gamma^t r_t \mid P, s_0, \pi]$. Note that the size of the tokenized state and action spaces grow exponentially with the size of the vocabulary. This makes it intractable to solve the MDP even for a medium-size vocabulary. As a result, it would quite desirable to develop a novel MDP paradigm that is more amendable to RL-based DM systems.

## 3 MIXTURE OF EXPERTS (MoE) LANGUAGE MODEL

We start by explaining how a MoE language model (MoE-LM) can enrich the bot's utterances and improve the overall performance of the DM. While our approach is applicable to any DM system, we use open-domain dialogue (Sankar et al., 2019) as a running example to show how MoE-LM-based agents can improve user satisfaction measured by an improvement on a sentiment or engagement. Intuitively a good DM agent should possess different behaviors (e.g., inquisitive, explorative, relevant, soothing, empathetic, complimentary, provoking) and swiftly decide which intent to use to pivot a conversation, build rapport, pique the user's interests, improve their mood, etc. To achieve this goal, we require the LM to have a language representation (primitive discovery) that captures different semantics, in order to encode different conversations and avoid generating dull and repetitive responses. We also need a machinery (expert construction) to embed different intents into sub-models of this LM, so that they can behave accordingly when prompted, and respond efficiently. Finally, with various candidate utterances available, the DM module of this LM should understand the current level of user satisfaction and determine which response is the most appropriate. Motivated by these observations, we construct our MoE-LM in three steps as shown in Figure 1. We give the main idea behind each step here and leave their detailed descriptions to Sections 4, 5, and 6.

**Step 1: Primitive Discovery.** We first employ the dataset $\mathcal{D}$, introduced in Section 2, and learn a language model $\texttt{LM}_0 = (\Phi, \mathcal{G}_0, \Psi)$ consisting of a *stochastic encoder* (i.e., an encoder $\Phi$ and a latent space distribution $\mathcal{G}_0$ that maps the encoded conversation into a latent distribution), and a decoder $\Psi$. The stochastic encoder $(\Phi, \mathcal{G}_0)$ comprises an encoder $\Phi$ that maps tokenized conversation histories $\mathbf{X}$ to a latent space $\mathcal{Z} \subseteq \mathbb{R}^d$, i.e., $z = \Phi(\mathbf{X}) \in \mathcal{Z}$, which is then used to construct a parameterized $d$-

---

[2]Note that we use $Y$ as the next utterance in the dataset and $\hat{Y}$ as the one predicted by the LM.

dimensional Gaussian distribution $\mathcal{G}_0(z'|z) = \mathcal{N}\big(\mu_0(z), \sigma_0^2(z)\mathbf{I}_{d\times d}\big)$ over $\mathbb{R}^d$. The decoder predicts the next utterance $\widehat{Y}_0$ (token-by-token) conditioned on the point $z'$ sampled from the latent distribution, i.e., $\Psi(\widehat{Y}_0|z')$[3], $z' \sim \mathcal{G}_0(\cdot|z)$. We denote by $\texttt{LM}_0(Y|\mathbf{X}) := \mathbb{E}_{z'\sim\mathcal{G}_0(\cdot|z),z=\Phi(\mathbf{X})}[\Psi(Y|z')]$, the *primitive* and learn it using a loss function that in addition to predicting the next utterance accurately, encourages diversity and generalization in the learned latent space $\mathcal{Z}$ (see Eq. 1 and Fig. 2). As we will explain in Section 4, our loss function is inspired by those in prior work, and more specifically by the one in OPAL (Ajay et al., 2020), i.e., an unsupervised learning method for discovering primitive skills in trajectories that are used by some downstream RL tasks.

**Step 2: Expert Construction.** Given the latent space $\mathcal{Z}$, encoder $(\Phi, \mathcal{G}_0)$, and decoder $\Psi$ learned in Step 1, we now learn $m$ latent distributions $\{\mathcal{G}_i\}_{i=1}^m$, each defined as $\mathcal{G}_i(z'|z) = \mathcal{N}\big(\mu_i(z), \sigma_i^2(z)\mathbf{I}_{d\times d}\big)$. Intuitively, each $\mathcal{G}_i$ corresponds to an attribute, e.g., an intent or a personality (in case of a chatbot) and generates samples in specific parts of the latent space $\mathcal{Z}$. This results in having $m$ LMs, $\{\texttt{LM}_i\}_{i=1}^m$, $\texttt{LM}_i = (\Phi, \mathcal{G}_i, \Psi)$, each of them corresponds to a *specialized* version of the original LM, $\texttt{LM}_0$, and serves as an *expert* in our MoE-LM. Upon receiving a conversation history $\mathbf{X}$, each expert $\texttt{LM}_i$ generates a candidate (or more) for the next utterance $\widehat{Y}_i$ in certain parts of the language space that are compatible with its attribute (personality). As we will explain in Section 5, each $\mathcal{G}_i$ is learned using a loss function that encourages its corresponding LM, $\texttt{LM}_i$, to generate utterances consistent with its attribute (see Eq. 2).

**Step 3: Dialogue Manager (DM).** The *dialogue manager*, denoted by $\mu$, takes as input the encoded conversation history $z = \Phi(\mathbf{X})$ and the candidate action utterances generated by the experts $\{\widehat{Y}_i\}_{i=0}^m$, and selects one of them as the action for the bot to execute, i.e., $\widehat{Y} \sim \mu(\cdot \mid z, \{\widehat{Y}_i\}_{i=0}^m)$. We will describe how DM is trained using reinforcement learning (RL) in Section 6.

## 4 PRIMITIVE DISCOVERY IN MOE-LM

Motivated by literature in the reinforcement and imitation learning fields (Ajay et al., 2020), we propose to learn the primitive LM, $\texttt{LM}_0$, in our MoE-LM by solving the following KL-constrained optimization problem that aims at capturing diverse semantics:

$$\min_{(\Phi,\mathcal{G}_0,\Psi),\rho} \widehat{\mathbb{E}}_{z'\sim\rho(\cdot|z,Y),z=\Phi(\mathbf{X})}\big[-\log\Psi(Y|z')\big], \quad \text{s.t. } \widehat{\mathbb{E}}_{z=\Phi(\mathbf{X})}\big[\text{KL}\big(\rho(z'|z,Y) \,\|\, \mathcal{G}_0(z'|z)\big)\big] \le \epsilon_{\text{KL}}, \quad (1)$$

where $\widehat{\mathbb{E}}$ is the empirical expectation over $(\mathbf{X}, Y)$ in the dataset $\mathcal{D}$, $\rho$ is a distribution over the latent space conditioned on the encoded conversation history $z$ and the target utterance $Y$, and $\epsilon_{\text{KL}}$ is a positive real-valued threshold. Using (1), we learn $\texttt{LM}_0 = (\Phi, \mathcal{G}_0, \Psi)$ by maximizing the log-likelihood, while enforcing consistency between the latent variable $z'$ predicted by $\mathcal{G}_0(\cdot|z)$ and $\rho(\cdot|z, Y)$ via the KL constraint. The distribution $\rho(\cdot|z, Y)$ is a Gaussian $\mathcal{N}\big(\mu_\rho(z, \Phi_\rho(Y)), \sigma_\rho^2(z, \Phi_\rho(Y))\mathbf{I}_{d\times d}\big)$ in which $\Phi_\rho$ is a pre-trained encoder for the target utterance $Y$, and mean $\mu_\rho(\cdot, \cdot)$ and variance $\sigma_\rho^2(\cdot, \cdot)$ are trainable models. One reason for using a separate encoder $\Phi_\rho$ for the target utterance $Y$ is to avoid overfitting $\Phi$ (i.e., to avoid having back-propagation gradient of $\Phi$ with $Y$ as input).

**Connection to VAE-like objectives** In practice, we implement the KL constraint in (1) as a penalty weighted by an appropriately chosen coefficient. Thus, one may interpret the objective in (1) as a variation of $\beta$-VAE (Burgess et al., 2018). Due to the connection to VAEs, one may draw similarities between our method and existing dialogue approaches such as VHRED (Serban et al., 2017b) and VHCR (Park et al., 2018). However, we emphasize that there are key differences, and these may be explained by first understanding how the objective in (1) encourages *diversity*, which is key to good primitive learning. Namely, it is important that primitive discovery learns an encoder-decoder $\Phi, \Psi$ which can be modulated by the choice of $z$; i.e., changing $z'$ while fixing $\mathbf{X}$ should lead to different distributions over generated utterances. The objective in (1) encourages this diversity by conditioning the latent variable $z'$ on both the target utterance $Y$ and $z = \Phi(\mathbf{X})$, i.e., $z' \sim \rho(\cdot|z, Y)$. In contrast, the KL constraint is used to make sure that the stochastic encoder $\mathcal{G}_0(\cdot|z)$ of our primitive LM is not too varied for different $Y$, and thus has a *limiting* effect on diversity. For example, in the extreme when $\epsilon_{\text{KL}} = 0$ (or, $\beta \to \infty$ when used as a regularizer) there will be no specialization of the latent space for different $Y$. Although $\beta \to \infty$ is an extreme case, degenerate behavior can also happen when $\beta = 1$, i.e., in the traditional variational loss used by VHRED and VHCR. Specifically, it is well-known that the traditional VAE loss is an upper bound on the negative log-likelihood of the data under a stochastic encoder-decoder parameterization. Thus if the data can be modeled by a single

---

[3]In practice, we can use both latent states as the input to the decoder model $\Psi(\widehat{Y}_0|z', z)$.

LM then a VAE-optimal decoder $\Psi$ can simply ignore $\mathcal{G}_0$, leading to a degenerated latent space as observed in previous work (Park et al., 2018). This is precisely the reason that, in our approach, we weaken the KL constraint ($\epsilon_{\text{KL}} \gg 0$ or, equivalently, $\beta \ll 1$). This enables our approach to more reliably guarantee that a unique $z'$ represents each distinct conversation pair $(\mathbf{X}, Y)$, thus capturing diverse semantic modalities and enabling easier downstream specialization.

In the mathematical results below, we formalize the claim above, namely, that the log-likelihood objective in (1) leads to a learned $\Phi, \Psi$ that can easily recover any arbitrary desired LM by specializing the latent space $\mathcal{G}$. We begin with a definition that characterizes the coverage of an arbitrary LM on the conditional conversation data distribution $P_{\mathcal{D}}(Y|\mathbf{X})$.

**Definition 1.** $\text{LM}_{\mathcal{D}, \xi}$ is a $\xi$-common LM of data $\mathcal{D}$ if $\mathbb{E}_{\mathcal{D}}[\text{TV}(\text{LM}_{\mathcal{D}, \xi}(Y|\mathbf{X}) || P_{\mathcal{D}}(Y|\mathbf{X})))] \leq \xi$.

Leveraging Theorem 4.1 in Ajay et al. (2020), we now present the theoretical result characterizing the representational power of our primitive encoder-decoder pair $(\Phi, \Psi)$ on data $\mathcal{D}$.

**Lemma 1.** Let $(\Phi, \rho, \Psi)$ be the solution to (1) with $\widehat{\mathbb{E}}_{z' \sim \rho(\cdot|z, Y), z = \Phi(\mathbf{X})}[-\log \Psi(Y|z')] = \epsilon$. Then there exists $\text{LM} := (\Phi, \mathcal{G}, \Psi)$ such that $\mathbb{E}_{\mathcal{D}}[\text{TV}(\text{LM}_{\mathcal{D}, \xi}(Y|\mathbf{X}) || \text{LM}(Y|\mathbf{X}))] \leq \xi + \sqrt{\frac{1}{2}(\epsilon + \mathcal{H})}$, where $\mathcal{G}(z'|z) = \mathbb{E}_{Y \sim \mathcal{D}}[\rho(z'|z, Y)]$, and $\mathcal{H} = \mathbb{E}_{\mathcal{D}}[\log P_{\mathcal{D}}(Y|\mathbf{X})]$ is a constant depending on $\mathcal{D}$.

The result above shows that, as long as $\text{LM}_{\mathcal{D}, \xi}$ is $\xi$-common in $\mathcal{D}$, then there exists a specialization of the latent space $\mathcal{G}$ that, when paired with $\Phi, \Psi$, can approximately recover $\text{LM}_{\mathcal{D}, \xi}$. The quality of the approximation is a function of $\epsilon$ — how well the objective in (1) was optimized — and $\xi$. In practice, to construct the primitive by replacing $\mathcal{G}$ with $\mathcal{G}_0$, i.e., $\text{LM}_0 = (\Phi, \mathcal{G}_0, \Psi)$, because $\mathcal{G}_0(z'|z)$ can be viewed as an distillation of $\rho(z'|z, Y)$. This theoretical result also motivates the next section, where we explain our algorithm's "Step 2: Expert Construction". Specifically, we show how to use the trained encoder-decoder pair $\Phi, \Psi$ to learn a spectrum of different specialized experts parameterized by different latent distributions $\mathcal{G}_i$.

## 5 EXPERT CONSTRUCTION WITH PLUG-AND-PLAY LANGUAGE MODELS

To complete the MoE framework one needs to systematically create a gamut of different experts $\text{LM}_i, \forall i \in \{1, \ldots, m\}$, with each generating candidate utterances of different intents. By viewing each expert as a distribution of particular behaviors in conversation data $\mathcal{D}$, we leverage the results of Section 4 and Lemma 1 and adopt a universal encoder-decoder $(\Phi, \Psi)$ among all the experts. Therefore, each expert $i$ is only parameterized by an arbitrary $d$-dimensional latent distribution (e.g., Gaussian), and it samples certain regions of the latent space $\mathcal{Z}$. Following the terminology from Dathathri et al. (2019), these experts can all be catagorized as *plug-and-play language models (PPLMs)*. Creating experts is handy because it only requires learning new latent distributions, while switching between experts amounts to sampling a different distribution.

Denote by $\ell_i(\mathbf{X}, Y) \in \mathbb{R}$ a real-valued label that *characterizes* the intent of expert $i \in \{1, \ldots, m\}$, e.g., determined by an off-the-shelf sentiment classifier. We train the latent distribution $\mathcal{G}_i(z)$ of expert $i$ by solving the optimization problem

$$\min_{\mathcal{G}_i} \widehat{\mathbb{E}}_{z' \sim \mathcal{G}_i(\cdot|z), z = \Phi(\mathbf{X}), Y \sim \Psi(\cdot|z')}[-\ell_i(\mathbf{X}, Y)]. \tag{2}$$

Unlike the weighted maximum likelihood approach considered in Dathathri et al. (2019), which assigns weight $\ell_i$ to training samples that correspond to expert $i$, we propose to learn each expert via *reward-maximization* and treat $\ell_i$ as a reward signal w.r.t. expert $i$ to be maximized. Interestingly, this approach is also linked to reinforcement learning (RL), in which both the "state" and "action" spaces are the latent space $\mathcal{Z}$, and the "policy" is the latent distribution $\mathcal{G}_i$. The main benefit of our approach is that it does not require the target utterance $Y$ from data $\mathcal{D}$ and is thus less vulnerable to data-imbalance issues in $\mathcal{D}$ on certain intents. Notice from (2) that the reward-maximization problem is myopic, i.e., the above RL problem has a discounting factor of 0. The main motivation is that, unlike dialogue management that is a sequential decision-making problem, here we want each expert to possess particular behaviors, and this can readily be done via greedy maximization. Long-term dialogue optimization will be handled by the dialogue manager rather than these experts.

For example in the case of a Gaussian $\mathcal{G}_i$, we use the standard REINFORCE (Sutton et al., 1999) algorithm to learn the model parameters $(\mu_i, \sigma_i^2)$ of $\mathcal{G}_i$ according to

$$\{\mu_i, \sigma_i\} \leftarrow \{\mu_i, \sigma_i\} + \alpha \cdot \mathbb{E}_{z' \sim \mathcal{G}_i(\cdot|z), Y \sim \Psi(\cdot|z')}[\ell_i(\mathbf{X}, Y) \cdot \nabla_{\{\mu_i, \sigma_i\}} \log \mathbb{P}_{\mathcal{G}_i}(z'|z)], \; i \in \{1, \ldots, m\},$$

where $\alpha > 0$ is the learning rate. To reduce the variance of these estimates, we also adopt the baseline reduction technique (Greensmith et al., 2004) in policy gradient, which replaces $\ell_i(\mathbf{X}, Y)$

with $\bar{\ell}_i(\mathbf{X}, Y) := \ell_i(\mathbf{X}, Y) - \mathbb{E}_{Y \sim \Psi(\cdot | \Phi(\mathbf{X}))}[\ell_i(\mathbf{X}, Y)]$. Following arguments from Lemma 1 and Lemma 4.0.1 in Ajay et al. (2020), in the following we quantify the sub-optimality of expert $\text{LM}_i$.

**Corollary 1.** Denote the $i$-th reward-maximizing objective as $\mathcal{L}_i(\text{LM}) := \widehat{\mathbb{E}}_{Y \sim \text{LM}(\cdot | \mathbf{X})}[\ell_i(\mathbf{X}, Y)]$. Suppose an optimal LM for this objective $\text{LM}_{i,\xi} \in \arg\max_{\text{LM}} \mathcal{L}_i(\text{LM})$ is $\xi$-common in $\mathcal{D}$. Moreover, let $\mathcal{G}_i^\star$ be in the $\arg\min$ of (2). Then with expert $\text{LM}_i = (\Phi, \mathcal{G}_i^\star, \Psi)$ and $(\epsilon, \mathcal{H})$ from Lemma 1, we have $|\mathcal{L}_i(\text{LM}_i) - \mathcal{L}_i(\text{LM}_{i,\xi})| \leq 2\|\ell_i\|_\infty \cdot (\xi + \sqrt{\frac{1}{2}(\epsilon + \mathcal{H})})$.

While it may be obvious that optimizing $\mathcal{G}_i$ w.r.t. (2) encourages expert $\text{LM}_i$ to capture the behaviors encouraged by $\ell_i$, this corollary has two further implications: (i) Since the sub-optimality of $\text{LM}_i$ compared to the oracle $\text{LM}_{i,\xi}$ is bounded by the quantity $\epsilon$ defined in Lemma 1, it justifies using the primitive $(\Psi, \Phi)$, which optimizes $\epsilon$, for expert construction; (ii) Sub-optimality further depends on $\xi$, quantifying how well $\text{LM}_{i,\xi}$ is represented in the original dataset $\mathcal{D}$.

## 6 REINFORCEMENT LEARNING FOR MoE-LM DIALOGUE MANAGER

We now describe the dialogue manager (DM) of our MoE-LM and propose RL algorithms to train it. As mentioned in Section 3, the DM is a policy $\mu$ that takes the encoded conversation history $z = \Phi(\mathbf{X})$ and the $m + 1$ candidate action utterances generated by the experts $\{\widehat{Y}_i\}_{i=0}^m$,[4] and stochastically selects one of them to execute, i.e., $\widehat{Y} \sim \mu(\cdot \mid z, \{\widehat{Y}_i\}_{i=0}^m)$. Note that each expert $i \in \{0, \ldots, m\}$ is a LM, $\text{LM}_i$, that acts as a policy $\pi_i(\cdot | \mathbf{X})$ and maps each conversation history $\mathcal{X}$ to an utterance $\widehat{Y}_i$. With this architecture we address the large size of state and action spaces in the original MDP that grows exponentially with the size of the vocabulary. As described in Section 2, the state and action spaces of the original MDP are the tokenized conversation history and the tokenized language space, respectively, while here the DM should choose among $m + 1$ actions (which is a much smaller and finite action space) given the latent space $\mathcal{Z}$ (which is a continuous state space) of encoded conversations. It is important to note that our MoE-LM is different than other hierarchical architectures (Kulkarni et al., 2016) in which the decision at the high-level is to choose a low-level controller only based on the current state of the system. In MoE-LM, the DM observes both the current state and the actions suggested by the experts and then chooses one among them.

We consider two RL settings to solve this specialized MDP. The first one is offline RL, in which the policy must be learned from the collected conversations $\mathcal{D}$ without further (online) interactions. Offline RL requires optimizing a policy, while minimizing the deviation from the behavior policy to avoid errors due to data co-variate shift. Among numerous offline RL algorithms (Kumar et al., 2020; Carta et al., 2021; Jaques et al., 2019), one effective algorithm to learn the DM policy $\mu$ is IQL (Kostrikov et al., 2021). Given conversation data $\mathcal{D}$, IQL first computes the critic functions $(Q_{\theta^*}(z, a), V_{\phi^*}(z))$ via solving $\min_{\theta, \phi} \mathbb{E}_{(z,a,r,z_+) \in \mathcal{D}}[(r + \gamma V_\phi(z_+) - Q_\theta(z, a))^2] + \lambda \cdot \mathbb{E}_{(z,a) \in \mathcal{D}}[L_2^\tau(Q_\theta(z, a) - V_\phi(z))]$, where $z$ is the encoded conversation, $a$ is the bot utterance, $z_+$ is the next encoded conversation, $r$ is the conversation-level reward, $\lambda > 0$ is a tunable weight, and $L_2^\tau$ is the expectile regression operator (Koenker and Hallock, 2001) of estimating the top-$\tau$ expectile statistics of the $Q$-function random variable (approximated by the value function $V_\phi$), and then extracts the DM policy $\mu$ via greedification over the finite set of MoE candidate utterances: $\mu(a \mid z, \{\widehat{Y}_i\}_{i=0}^m) = \arg\max_{a \in \{\widehat{Y}_i\}_{i=0}^m} Q_{\theta^*}(z, a)$. Intuitively, IQL leverages the generalization capacity of critic functions to estimate the value of the best action without directly querying the values w.r.t. unseen actions. This makes it less conservative than most offline RL methods that either constrain the policy's actions to be in-distribution or solve a behavior-regularized policy optimization problem.

The second RL setting for learning the DM policy $\mu$ is via model-based RL (MBRL) (Shah et al., 2018; Wei et al., 2018). While this paradigm can be applied to any online/offline RL algorithms we demonstrate it with the simple DQN (Mnih et al., 2013). Here we first learn a user utterance model $P_{\text{user}}(X_+ | \mathbf{X}, a) := \mathbb{E}_{z = \Phi_{\text{user}}([\mathbf{X}, a])}[\Psi_{\text{user}}(X_+ | z)]$ via maximum likelihood, then generate data $\mathcal{D}_{\text{MB}}$, whose next-state $\widehat{s}_+$ encodes the next conversation generated from roll-outs and the corresponding candidate actions, solve the Bellman error minimization: $\min_\theta \sum_{(s,a,r,\widehat{s}_+) \in \mathcal{D}_{\text{MB}}}(\bar{r} + \gamma Q_{\theta^{\text{target}}}(\widehat{s}_+, \arg\max_{a_+ \in \{0,\ldots,m\}} Q_\theta(\widehat{s}_+, a_+)) - Q_\theta(s, a))^2$, and obtain the DM policy $\mu$ via the aforementioned greedification step. The benefit of MBRL over offline RL is two-fold. First, one can easily obtain a *high-fidelity* user utterance model (Peng et al., 2020) by simply fine-tuning a large LM (e.g., GPT-3 (Floridi and Chiriatti, 2020)). Second, with sufficient

---

[4]For simplicity, we assume that each expert generates only a single candidate utterance at each step. It would be straightforward to extend this to multiple (and even a varying number of) candidate utterances.

dialogue roll-outs that captures many different scenarios, MBRL generally can be more data-efficient and less prone to distributional shifts than offline RL.

## 7 EXPERIMENTS

We evaluate our MoE-approach on two open-domain benchmarks that are common within the RL-based dialogue management literature (Jaques et al., 2019). The first one is the Cornell Movie corpus (Danescu-Niculescu-Mizil and Lee, 2011), which consists of conversations between speakers in different movie lines and has a median conversation length of 3 utterances. The second is the Reddit Casual (Ghandeharioun et al., 2019) conversations, which is a subset of the Reddit corpus that only contains casual conversations on various topics of at least 3 turns and a median of 7 utterances.

We conduct several experiments to test the efficacy of different parts in the MoE-LM, namely (i) the predictive power and diversity of the primitive, (ii) the quality of experts, and (iii) the overall DM performance. For each metric, we report mean $\pm$ standard error over $100$ conversations sampled from the evaluation set. We also ran an ablation study on $4$ transformer-based MoE-LMs, namely MoE-1, MoE-2, MoE-3- MoE-4, to understand how performance is affected by different model architectures, language encoders, and latent generators. MoE-1 and MoE-2 use a simpler architecture, while MoE-3 and MoE-4 use the same encoder architecture as BERT (Devlin et al., 2018). MoE-1 uses much smaller latent distribution models $\{\mathcal{G}_i\}$ than MoE-2; MoE-3 uses the pre-trained BERT encoder, while MoE-4 trains that from scratch. Details of these models can be found in Appendix B.3.

**EXP 1: Comparing Primitive Models** We compare the quality of latent representations learned by the 4 MoE-LMs (via optimizing Eq. 1) and 2 baselines (standard Transformer (Vaswani et al., 2017) and VHRED [5] (Serban et al., 2017b)). To assess their quality, for each test conversation we generated 25 utterances and reported the following 3 metrics: (i) **Diversity**: The $1-$ Sparsity (Hurley and Rickard, 2009) of the singular values of the embedded utterances, i.e., $\text{Diversity}(\{\hat{Y}_i\}) := 1 - \sqrt{d - \|\text{SVD}\|_1/\|\text{SVD}\|_2}/\sqrt{d-1} \in [0,1]$, where $\text{SVD} := \text{SVD}(\{\Phi_{\text{SE}}(\hat{Y}_i)\}_{i=1}^{25})$, and $\Phi_{\text{SE}}$ is a pre-trained sentence encoder (e.g., a USE (Cer et al., 2018)); (ii) **Dist-$\{1, 2, 3\}$** (Li et al., 2015): Ratio of unique $\{1, 2, 3\}$-gram in the generated utterances; (iii) **Perplexity**: The perplexity score of the utterance w.r.t. a GPT-2 LM, which is more correlated to human's judgement on semantic fluency (Pang et al., 2020). These metrics measure both accuracy and semantic diversity. Qualitatively, we also measure fluency and diversity of LMs using human ratings (see Appendix B.8 for details). The results of the above experiments are reported in Table 1 and 6 (Appendix A.1), and sample utterances generated by these LMs can be found in Appendix A.3. Human evaluation on diversity/fluency of different LMs are given in Table 4. In comparisons with the baselines (Transformer and VHRED), generally (i) transformer-based LMs out-perform VHRED due to their attention mechanism that explicitly encodes sequential semantic information, and (ii) the MoE-LMs achieve way better diversity without sacrificing much on accuracy (i.e., the perplexity scores are still quite low). Qualitatively, the sample utterances generated the Transformer are closer to the targets than that by MoE-2 and MoE-4, likely because Transformer tends to memorize the corpus (Kharitonov et al., 2021). Contrarily, MoE-LMs generate utterances that have similar contexts with targets but paraphrased or similar structures but different contexts, demonstrating their generalizability. Human evaluations also show that MoE-2 and MoE-4 generate more diverse utterances while retaining sufficient semantic fluency.

Among different MoE-LMs, MoE-2 and MoE-4 have the best performances, particularly MoE-4 has better diversity while MoE-2 has lower perplexity. This corroborates with our hypotheses that (i) jointly training the encoder and decoder with Eq. 1 seems necessary to encourage semantic diversity (as opposed to using a pre-trained BERT encoder, which maximizes likelihood), (ii) sufficient representation power is necessary for $\mathcal{G}_0$ to match the posterior distribution $\rho$ in order to capture different semantics in $\mathcal{D}$. In Fig. 2a and 2b, we visualize the latent space of $400$ conversation data samples for both Transformer and MoE-4. The latent states of MoE-4 are much more dispersed across the embedding space, implying that most conversations get encoded uniquely. In contrast, the latent space of Transformer has many clusters, suggesting it is more prone to generating similar utterances even with different input conversation and is thus less generalizable.

**EXP 2: Quality of Experts** We compare the performance of experts learned by the 4 MoE-LMs (where experts are separately trained by optimizing Eq. 2) and 2 baselines (WD (Holtzman et al., 2018) and PPLM (Dathathri et al., 2019)). To study the sub-optimality gap in Corollary 1, we also include the performance of Transformer-based expert end-to-end LMs that are individually optimized with REINFORCE (Li et al., 2016), using the expert labels $\{\ell_i\}$ as rewards. Inspired by Ghandeharioun et al. (2019) on how bot behaviors are characterized, we use the following label functions to define the

---

[5] The VHRED model implementation is identical to that in Jaques et al. (2019) to ensure fair comparisons.

| Method | Diversity | Dist-1 | Dist-2 | Dist-3 | Perplexity |
|---|---|---|---|---|---|
| MoE-1 | $0.069 \pm 0.03$ | 0.27 | 0.66 | 0.75 | $27.43 \pm 18.49$ |
| MoE-2 | $0.14 \pm 0.05$ | 0.35 | 0.77 | 0.90 | $38.81 \pm 17.34$ |
| MoE-3 | $0.089 \pm 0.04$ | 0.29 | 0.75 | 0.90 | $41.35 \pm 26.68$ |
| MoE-4 | $\mathbf{0.16 \pm 0.04}$ | **0.38** | **0.80** | **0.95** | $50.17 \pm 28.11$ |
| Trans. | $0.087 \pm 0.03$ | 0.26 | 0.65 | 0.85 | $\mathbf{19.23 \pm 15.46}$ |
| VHRED | $0.09 \pm 0.04$ | 0.35 | 0.70 | 0.79 | $79.77 \pm 39.61$ |

Table 1: Accuracy (Perplexity) and Diversity of Language Primitive Experts Trained with Reddit.

| Method | User Tot. Sent. | User Sent. Trans. | Perplexity |
|---|---|---|---|
| MoE-4 IQL | $\mathbf{4.55 \pm 0.38}$ | $\mathbf{2.88 \pm 0.35}$ | $45.53 \pm 26.71$ |
| MoE-4 Ens-Q | $4.14 \pm 0.41$ | $2.36 \pm 0.31$ | $43.77 \pm 28.24$ |
| MoE-4 KLC | $3.94 \pm 0.25$ | $2.21 \pm 0.24$ | $\mathbf{38.35 \pm 16.88}$ |
| VHRL | $3.85 \pm 0.28$ | $2.19 \pm 0.28$ | $55.81 \pm 24.21$ |
| VHRL-KLC | $3.95 \pm 0.19$ | $2.16 \pm 0.33$ | $64.05 \pm 36.98$ |
| VHRL-SAC | $3.93 \pm 0.28$ | $2.19 \pm 0.32$ | $62.06 \pm 40.43$ |

Table 2: Performance (w.r.t. User Satisfaction in Conversation) of MBRL-based DM Trained with Reddit.

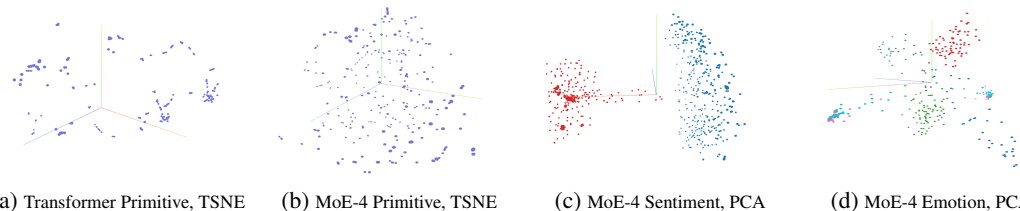

(a) Transformer Primitive, TSNE    (b) MoE-4 Primitive, TSNE    (c) MoE-4 Sentiment, PCA    (d) MoE-4 Emotion, PCA

Figure 2: Latent Space Visualizations. Figures (a) and (b) Compares Two Primitive Representations. Figures (c) and (d) Illustrates How Experts (of Different Sentiments and Emotions) are Represented by Latent Clusters.

intents of experts: (i) $\ell_{\text{pos-sent}}(Y)$, $\ell_{\text{neg-sent}}(Y)$, $\ell_{\text{joy}}(Y)$, $\ell_{\text{optimism}}(Y)$, $\ell_{\text{anger}}(Y)$, $\ell_{\text{sadness}}(Y)$ quantify 6 different sentiment tones and are constructed by a RoBERTa-based sentiment detector (Liao et al., 2021) that predicts whether an utterance is of positive or negative sentiment, and whether it falls into any of the 4 more-refined emotions: {joy, optimisim, sadness, anger}; (ii) $\ell_{\text{sent-coh}}(\mathbf{X}, Y)$ measures *empathy*, i.e., bot's sentiment coherence with user's; (iii) $\ell_{\text{question}}(Y)$ outputs 1 when a bot question is detected and 0 otherwise; (iv) $\ell_{\text{exp}}(\mathbf{X}, Y)$ quantifies *exploration*, i.e., the tendency to avoid repetitive contexts (see Appendix B.7 for details). Qualitatively, we also measure fluency and expert skills of LMs using human ratings (see Appendix B.8 for details). The results of the above experiments are reported in Table 3 and 8 (Appendix A.1), with sample utterances reported in Appendix A.4 to A.10. Results on human evaluation of different LMs w.r.t. fluency and different expert skills are given in Table 5. Compared with the baseline LMs, generally the experts created under the MoE-LM framework, especially under MoE-2 and MoE-4, better capture all different language intents (where WD and PPLM appear to capture negative sentiments and emotions much more effectively than behaviors), demonstrating the efficacy of our approach which constructs specialized experts on a diverse language space via reward maximization (instead of weighted MLE). Human evaluations also show that MoE-4 is most effective in generating semantically fluent utterances that possess a wide range of expert characteristics.

Similar to the ablation study in EXP 1, all the experts associated with MoE-2 and MoE-4 are also among the best ones in capturing language intents. Interestingly, with the Reddit data the experts in MoE-4 perform the best, while with much less data (Cornell) the best experts are built upon the simpler MoE-2 architecture. We conjecture this difference is due to over-fitting issues faced by the larger LMs (MoE-4) when there is insufficient data for expert fine-tuning. In Fig. 2c and 2d we visualize the latent space of the sentiment-based experts in MoE-4, each with 400 conversation data samples. Notice that the sentiment experts' latent distributions are clearly separated (because positive and negative sentiments have opposite behaviors), while the emotion expert's latent distribution have more gradual separations and even some overlaps (because e.g., joy versus optimism are quite similar, while joy versus anger are quite different). This validates our MoE-LM represents different behaviors in separate regions of the latent space and justifies our structural prior of modeling each expert as a specialized version of the primitive LM, whose latent distribution focuses on particular regions.

**EXP 3: MoE-RL Against DialoGPT Simulated Users** We compare the dialogue management performance of MoE-LM, for which their MoE-based DMs $\mu$ are trained with different methods: (i) IQL (Kostrikov et al., 2021), (ii) Ensemble DQN (Carta et al., 2021), (iii) KL-control (Jaques et al., 2019), with 3 standard RL-based DM baselines using the VHRL architecture (Saleh et al., 2020): (i) REINFORCE (Li et al., 2016), (ii) KL-control, (iii) SAC (Haarnoja et al., 2018). According to the results on expert quality in EXP2, we pick the MoE-2 and MoE-4 frameworks for the Cornell and Reddit tasks respectively. For systematic evaluation, we perform the experiment by having these RL agents interact with a DialoGPT (Zhang et al., 2019) simulated user environment for a maximum of 5 turns. The DM task is to maximize total user satisfaction in the conversation level, which is measured by both (i) user's overall sentiment, and (ii) user's sentiment transition. To construct an immediate reward that serves as a surrogate for user satisfaction, we set $r(\mathbf{X}, a, X_+) = \lambda_1 \ell_{\text{sent}}(X_+) + \lambda_2 (\ell_{\text{sent}}(X_+) - \frac{1-\gamma}{1-\gamma^L} \sum_{l=0}^{L-1} \gamma^l \ell_{\text{sent}}(X_l))$, where the linear combi-

| Method | Question | Exploration | Positive Sent. | Negative Sent. | Sent. Coherence | Joy | Optimism | Anger | Sadness |
|---|---|---|---|---|---|---|---|---|---|
| MoE-1 | $0.65 \pm 0.20$ | $0.45 \pm 0.17$ | $1.13 \pm 0.21$ | $0.35 \pm 0.19$ | $0.50 \pm 0.38$ | $0.96 \pm 0.26$ | $-0.21 \pm 0.56$ | $0.54 \pm 0.58$ | $0.99 \pm 0.83$ |
| MoE-2 | $0.95 \pm 0.27$ | $0.47 \pm 0.21$ | $3.29 \pm 0.33$ | $1.42 \pm 0.38$ | $0.51 \pm 0.40$ | $1.99 \pm 0.38$ | $1.25 \pm 0.43$ | $\mathbf{1.48 \pm 0.39}$ | $\mathbf{2.01 \pm 0.46}$ |
| MoE-3 | $0.41 \pm 0.35$ | $0.50 \pm 0.24$ | $1.23 \pm 0.78$ | $0.99 \pm 0.48$ | $\mathbf{0.66 \pm 0.35}$ | $1.02 \pm 0.29$ | $0.49 \pm 0.51$ | $0.53 \pm 0.49$ | $1.10 \pm 0.48$ |
| MoE-4 | $\mathbf{0.96 \pm 0.37}$ | $\mathbf{0.51 \pm 0.31}$ | $\mathbf{3.41 \pm 0.55}$ | $\mathbf{1.80 \pm 0.34}$ | $0.52 \pm 0.31$ | $\mathbf{2.05 \pm 0.55}$ | $\mathbf{1.57 \pm 0.44}$ | $1.42 \pm 0.42$ | $1.97 \pm 0.36$ |
| WD | $0.05 \pm 0.03$ | $0.15 \pm 0.37$ | $-0.50 \pm 0.74$ | $1.01 \pm 0.48$ | $0.51 \pm 0.20$ | $-0.51 \pm 0.39$ | $-0.84 \pm 0.76$ | $1.00 \pm 0.44$ | $1.27 \pm 0.67$ |
| PPLM | $0.20 \pm 0.25$ | $0.48 \pm 0.28$ | $0.44 \pm 0.41$ | $0.69 \pm 0.22$ | $0.53 \pm 0.31$ | $0.31 \pm 0.29$ | $0.40 \pm 0.55$ | $0.71 \pm 0.46$ | $0.98 \pm 0.59$ |
| Trans. RL* | $0.99 \pm 0.23$ | $0.54 \pm 0.18$ | $3.53 \pm 1.64$ | $1.89 \pm 1.20$ | $0.72 \pm 0.30$ | $2.88 \pm 0.96$ | $1.80 \pm 0.59$ | $1.62 \pm 0.75$ | $2.35 \pm 0.62$ |

Table 3: Quality of Each Expert PPLM Trained on Reddit Casual dataset w.r.t. its Trained Label. Trans. RL Corresponds to Individually Optimized LMs Using Expert Labels $\{\ell_i\}$ as Rewards.

| Method | Avg. Fluency | Diversity |
|---|---|---|
| MoE-1 | $0.72 \pm 0.02$ | $0.51 \pm 0.02$ |
| MoE-2 | $\mathbf{0.75 \pm 0.02}$ | $0.54 \pm 0.02$ |
| MoE-3 | $0.72 \pm 0.02$ | $0.42 \pm 0.03$ |
| MoE-4 | $0.65 \pm 0.03$ | $\mathbf{0.69 \pm 0.02}$ |
| Trans. | $0.70 \pm 0.02$ | $0.47 \pm 0.02$ |

Table 4: Phase 1 Raters Evaluation

| Method | Avg. Fluency | $S_{\text{Question}}$ | $S_{\text{Pos. Sent}}$ | $S_{\text{Neg. Sent}}$ | $S_{\text{Joy}}$ | $S_{\text{Anger}}$ |
|---|---|---|---|---|---|---|
| MoE-3 | $0.76 \pm 0.04$ | $0.27 \pm 0.05$ | $\mathbf{0.64 \pm 0.04}$ | $0.26 \pm 0.05$ | $0.41 \pm 0.04$ | $0.33 \pm 0.05$ |
| MoE-4 | $\mathbf{0.82 \pm 0.02}$ | $\mathbf{0.74 \pm 0.04}$ | $0.46 \pm 0.04$ | $\mathbf{0.44 \pm 0.03}$ | $\mathbf{0.51 \pm 0.04}$ | $\mathbf{0.43 \pm 0.03}$ |
| Trans. | $0.78 \pm 0.03$ | $0.12 \pm 0.03$ | $0.29 \pm 0.04$ | $0.19 \pm 0.04$ | $0.38 \pm 0.04$ | $0.33 \pm 0.05$ |

Table 5: Phase 2 Raters Evaluation (Reddit Casual Models).

nation weights $(\lambda_1, \lambda_2) = (0.75, 0.25)$ correlate with Ghandeharioun et al. (2019), and $\ell_{\text{sent}}(X)$ is the same RoBerTa-based sentiment labeler as in EXP2, which assigns a score from $[-1, 1]$ that is proportional to the positive sentiment and inversely proportional to the negative sentiment prediction probabilities. To ensure the baseline RL DM methods can also possess certain bot-level features, e.g., question, positive sentiment, etc., besides the above RL reward for user satisfaction we also optimize a linear combination of bot-based rewards when training the baseline models, see Appendix B of Saleh et al. (2020) for more details. Since the DM problem lasts at most 5 turns, we use this as the effective horizon and set $\gamma = 1 - 1/5 = 0.8$.

The results of the above experiments (performed in both offline RL and MBRL settings) are reported in Table 2, Table 7 (Appendix A.1), Table 9 and 10 (Appendix A.2), with sample utterances reported in Appendix A.11. Our experiments show that MoE-LMs outperform most baselines on DM performance. We attribute this finding to three factors: (i) MoE-LM restricts the action space into a smaller set of candidate utterances generated by experts (whose qualities are validated in EXP2), the corresponding RL problem then becomes simpler and requires less data (especially in Cornell) to solve; (ii) Unlike the baseline RL methods, which need to optimize both bot-and-user signals, the MoE DM agents focus on optimizing the user satisfaction goal and are therefore more effective; (iii) Among different RL settings, MBRL, which first learns a user utterance model (the model uses the same encoder from the primitive LM and learns a separate decoder for user-utterance prediction) then does DM, performs much better than offline RL methods that moderately improve upon the primitive LM (behavior policy). IQL-based dialogue managers are among the best across different settings potentially because IQL is more robust to co-variate shifts than standard RL methods, e.g., Ens-Q, SAC, and yet it is less conservative than the behavior-regularized algorithms, e.g., KLC. Interestingly, our MoE-LMs also have lower (better) perplexity scores than other methods. This may be due to the fact that MoE-LM uses pre-trained encoder and decoder from the primitive LM, which are optimized for generalization and accuracy, while other RL methods may distort their language representations to create utterances that maximize reward but become less natural.

## 8 CONCLUDING REMARKS

We developed a mixture-of-expert (MoE) approach for RL-based dialogue management (DM). Our MoE language model (MoE-LM) comprises of three main components: (i) a LM that can generate diverse semantics for conversation histories, (ii) a number of specialized LMs (or experts) that can produce utterances corresponding to a particular attribute or intent, and (iii) a RL-based DM that performs dialogue planning with the utterances generated by the experts. To understand how well our MoE approach generates diverse and sensible utterances, and solves DM problems, we evaluated it using two open-domain dialogue tasks and compared it with SOTA baselines. Our results showed that MoE-LM (i) improves diversity of text generation, (ii) can generate utterances with specific intents, and (iii) yields better overall performance. We consider our work as a step forward in creating steerable LMs that possess different intents and in training RL-based DMs that can carry on rich conversations. Future work includes improving the language representation with information-theoretic approaches, fine-tuning the experts based on the DM objective, extending the RL agent to track users' behaviors (via abstract belief states) and plan upon them, preventing RL dialogue agents from generating harmful behaviors (via enforcing safety constraints), and evaluating our MoE-LM on more realistic problems, such as information retrieval, recommendation, and negotiation.

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
