# OpenReview forum: "A Mixture-of-Expert Approach to RL-based Dialogue Management"
_ICLR.cc/2023/Conference — ICLR 2023 poster_

### Official Review · Reviewer_Esbb · 2022-10-18

**Confidence:** 4
**Correctness:** 4
**Technical Novelty And Significance:** 4
**Empirical Novelty And Significance:** 3
**Recommendation:** 8

**Clarity, Quality, Novelty And Reproducibility:**

There is a great deal packed into this paper and access to the appendices is essential to a full understanding.  In my opinion, the paper could be improved by reducing the content focussing on the core ideas and principal results.  I found it particularly difficult to understand the reward function and how it was used in training.  This is barely mentioned in section 6 and the definition in section 7 surely contains a typo.

**Strength And Weaknesses:**

The core of this paper is the idea of generating LM primitives using a modification of the scheme proposed by Ajay et al for accelerating off-line learning.  It uses a form of VAE to learn a distribution for the latent space which sits between the encoder and decoder and it uses REINFORCE to train the mapping functions to localise the latent space around regions corresponding to each personality trait.  The techniques used are (mostly) clearly explained and the approach is elegant and well-motivated.


**Summary Of The Paper:**

This paper address the problem of building an end-to-end trained dialogue system in which responses are coherent both in terms of semantics and also in terms of personality and sentiment.  The approach adopted is to use an encoder-decoder architecture in which the intermediate encoding is manipulated by a mapping function.  By having multiple mapping functions, each corresponding to a specific personality trait or sentiment, a family of language model experts is created all using the same encoder and decoder.  To generate responses, each LM generates its own distinct output and finally a dialogue manager selects one of the outputs using a policy trained to maximise user satisfaction.

The system is evaluated in terms of diversity and accuracy of the LM experts and the user satisfaction achieved by the overall system.  The VHRED system is used as a baseline and training uses the Cornell movie database and Reddit.  The dialog manager is tested against DialoGPT, and also using human raters.  The proposed system delivers the best performance on all of the main metrics.

**Summary Of The Review:**

The approach taken in this work is elegant and mathematically well motivated.  Overall it represents a useful step forward in creating LM-based dialogue models that can be steered to coherently convey differing emotions and personality traits.   In my view, it will stimulate further ideas in this import problem area.

---

### Official Review · Reviewer_FimV · 2022-10-24

**Confidence:** 4
**Correctness:** 3
**Technical Novelty And Significance:** 3
**Empirical Novelty And Significance:** 3
**Recommendation:** 6

**Clarity, Quality, Novelty And Reproducibility:**

It is a decent paper to employ the mixture of experts for the conversation response generation, with different experts to handle different aspects of the response generation, and the empirical experiments on two benchmarks support the motivation, with promising results.

**Strength And Weaknesses:**

Strength:
1. This paper proposes to use a mixture of experts language model (MoE-LM) to handle the different aspects of the dialog response generation.
2. It shows promising results on two benchmarks on the automatic evaluation.

Weakness:
1. It may lack the human evaluation to verify the effectiveness of the proposed method.

**Summary Of The Paper:**

This paper attacks the generic responses and user satisfaction problem in the dialog response generation, it proposes a RL-base dialogue management with a mixture of experts language model (MoE-LM), so that different experts can handle the different aspect, for example, diverse semantics, personality etc.; empirical experiments on two open-domain dialogue benchmarks show promising results.

**Summary Of The Review:**

The motivation of this work is to employ the mixture of experts language model to handle different aspects in the dialogue response generation, overall, it is a decent work to show the mixture of experts in the response generation, with promising empirical results on two open-domain response generation benchmarks; it would be better to add human evaluation to verify the effectiveness of the proposed method, to show it well aligned with the automatic evaluation.

---

### Official Review · Reviewer_ySba · 2022-10-28

**Confidence:** 3
**Correctness:** 1
**Technical Novelty And Significance:** 4
**Empirical Novelty And Significance:** Not applicable
**Recommendation:** 3

**Clarity, Quality, Novelty And Reproducibility:**

The paper has significant issues with clarity/readability:

- Experiment section is confusing and poorly structured. Please make it clear what evaluation metrics are used in each table, what conversations you are using for evaluation (and where they come from), whether you are performing automatic or human evaluation.
- Do not cite Wolf et al for Transformer. If you’re using a specific implementation of Transformer introduced by that paper, please make that clear.
- Why is the human evaluation in the appendix, while the main paper presents arbitrarily-defined rewards?
- You should not claim to be training on "Reddit" when only a small subset of the typical pushshift.io reddit data (used by DialoGPT, and other dialog models that claim to train on reddit) is used. The original paper should not have called it the reddit dataset. Please clarify this.

The approach in the paper is novel.

**Strength And Weaknesses:**

Strengths:
- Very interesting and novel idea. I hope that this approach is extended (e.g., to more appropriate datasets) and evaluated thoroughly.

Weaknesses:

My concerns all pertain to the experimentation which does not validate the claims made by this paper. Despite a promising and interesting method, this paper (1) compares to weak baselines, (2) relies on un-validated automatic metrics, (3) does a small inconclusive human evaluation against weak baselines, (4) on datasets that are not typically used.

- Why is the human evaluation in the appendix, while the main paper presents arbitrarily-defined rewards?
- You cannot define arbitrary metrics/rewards and present them as a meaningful comparison. There is a body of work on defining meaningful dialog evaluation metrics which measures correlations against human judgements (survey: https://arxiv.org/pdf/2106.03706.pdf). If you intend to define new automatic metrics, you must compare them to human judgments on benchmark corpora before using them.
- Human evaluation does not demonstrate a statistically significant improvement over the baseline transformer model. Furthermore, human evaluation should have been done against state-of-the-art dialog models.
- Human evaluation should present statistics like inter-annotator agreement, instructions to annotators, etc.


- The paper does not compare against any state-of-the-art dialog models. Why not compare to DialoGPT/Plato-2/BB2/BB3/etc? Responses in appendix seem subpar relative to typical responses produced by sota models.

- Evaluation datasets? Why reddit/cornell? Please justify this decision over using alternative, more common and well-accept dialog corpora (e.g., BST, Topical-Chat, DailyDialog).



**Summary Of The Paper:**

This paper introduces a Mixture of Experts Language Model (MoE-LM) that includes (1) a primitive LM that learns diverse semantics of dialog, (2) a set of expert LMs that generate responses according to a particular attribute (e.g., sentiment) and (3) an RL based DM that considers the dialog history and selects from the set of utterances generated by the experts. This paper is a promising proof of concept with a very interesting and novel idea -- however the experimentation is problematic and does not validate any of the claims of this paper

**Summary Of The Review:**

My concerns all pertain to the experimentation which does not validate the claims made by this paper. Despite a promising and interesting method, this paper (1) compares to weak baselines, (2) relies on un-validated automatic metrics (or metrics that are known to be faulty like Dist-1/2/3, LM PPL), (3) does a small inconclusive human evaluation against weak baselines, (4) on datasets that are not typically used.

As far as I understand (disclaimer: I did not look carefully into the math), the core idea of the paper is very promising. If the experimental issues are resolved (e.g., by following the typical experimental paradigm of sota dialog models), this will be a very strong paper.

---

### Official Review · Reviewer_8p8z · 2022-10-30

**Confidence:** 3
**Correctness:** 3
**Technical Novelty And Significance:** 3
**Empirical Novelty And Significance:** 3
**Recommendation:** 6

**Clarity, Quality, Novelty And Reproducibility:**

The technical novelty is presented very clearly in this paper. And to the best of my knowledge, the proposed model is novel. The technical details are clear, however, I think to reproduce the results of such a complex model, more training details should be presented, such as the choices of hyper-parameters and the training strategy. For example, has each LM be pretrained, and how they're pretrained? The code and training scripts should be released.

**Strength And Weaknesses:**

Strength:

1) The motivation is clear and compelling and the idea of changing the action space from word-level choices to utterance level is elegant.
2) From the perspective of technical contribution, the paper is very well written, and the math is clear and easy to follow.
3) The improvements in diversity compared to existing models are significant.

Weakness:

1) Although I like the math-centric writing of this paper, which defines all the problems very clearly and prevents confusion in technical details, I think this paper could be improved if you can cut some details in math (move them to the appendix) and save more spaces in more interpretable results analysis. After all, dialog generation is a human-centric task, and it's better to present more examples to show the real quality of the generated dialog. Moreover, since each LM is corresponding to a particular attribute, I think it's better to save some space to discuss how many attributes are defined and why are they chosen.

2) For evaluation, this paper spends quite some effort in comparing different types of LM architecture, I don't think it's necessary, the main contribution is the mixture model, it makes more sense to fix the architecture to a most common or a SOTA model in the single-LM RL-based DM approach, and focus the comparison in the improvements brought by the mixture model and how it affects the quality of the generated utterance.

3) Similarly to the previous question, I have some doubts about the choices of baseline. VHRED is a pretty ancient model whose LM is still LSTM. Why not comparing with the same type of LM?

4) I think human evaluation is necessary to demonstrate the actual qualtiy of the generated utterance. I can see some human evaluation is added in the supplementary materials. But they should be added in the main paper and should be well analyzed.

**Summary Of The Paper:**

In the existing application of language model (LM) in RL-based dialog management (DM) models, the agent is trained at the word-level and thus results in a complex action space. To address this issue, this paper proposes a novel mixture of expert language models (MoE-LM) and performs the planning at the utterance level instead of the word level. Each language model in the mixture models is specialized to generate utterances corresponding to a particular attribute or personality. Compared with SOTA baselines on open-domain dialogues demonstrate the effectiveness of the proposed MoE-LM in the diversity and sensibility of the generated utterances.

**Summary Of The Review:**

This paper has a strong technical contribution and the empirical results are significant. However, I think the paper can still be improved by setting up better baselines, including human evaluation and more examples. The details in training are also necessary to reproduce the results.

---

### Official Review · Reviewer_nxAX · 2022-11-03

**Confidence:** 3
**Correctness:** 4
**Technical Novelty And Significance:** 3
**Empirical Novelty And Significance:** Not applicable
**Recommendation:** 8

**Clarity, Quality, Novelty And Reproducibility:**

The paper is well written in terms of clarity. They note how one might think they are similar to other techniques like Hierarchical RL and VAEs and flesh out the differences that make them unique .
There have been other papers that use different subsets of these components or variations of components but this MoE-LM is novel in its optimization and combination of components.
They have listed all model parameters and choice of coefficients in detail in the appendix for reproducibility.


**Strength And Weaknesses:**

Strengths:

	1. They address many of the common problems faced by open domain dialogue namely diversity, modularity in experts , exploration, engagement.
	2. Each component can focus on different aspects by having their own optimization. The primitive LM's loss function focuses on generality and diversity, the experts, on greedy reward maximization of their intent and the dialogue manager optimizes on user sentiment/satisfaction as opposed to just incorporating all these factors in a linear fashion.
	3. They show how the experts show different behaviors in different regions of the latent space through this method
	4. RL optimization does not distort the encoder/decoder space learned by the LM making the utterances more robust

Weaknesses:

	1. They did not compare the whole system with other SOTA open domain dialog models like RetGen (the successor of DialoGPT), meena (Adiwardana et al., 2020), know-edg (Li et al., 2020a)
	2. They did not use the latest version of VHRED with linear Gaussian prior (Zhao and Kawahara 2020)  for evaluating their primitive language model.
	3. No back propagation of reward from dialogue management to language model.
	4. They did not tackle other problems in open domain dialogue such as memory to register and personalize to user characterstics  , reasoning over common sense and facts.
        5.Most of the evaluation details are in the appendix and not in the main paper


**Summary Of The Paper:**

This paper puts forth a new model for open domain dialogue which is a combination of RL based dialogue system and a mixture of expert language models trained on conversation histories. They explain in detail about the three components of this hybrid system : a primitive language model, expert plug and play transformer based language models for each intent (question, exploration, positive, negative, joy, optimism, anger, sadness) and an RL based dialogue manager , each with it's own loss function/reward function. They perform an ablation study with variations on the transformer models as well as a few RL techniques. They evaluate each component individually using automated metrics such as perplexity, diversity as well as human evaluation on fluency, diversity and intent precision and compared them to baselines.



**Summary Of The Review:**

There are several reason to accept this paper such as :

1. The effectiveness of the model shown through automated metrics as well as human evaluation .
2. The coverage of solutions to common open domain dialogue problems .
3. The ease with which experts can be trained with different intents and objectives and
4. The potential transfer of domain to task-oriented dialogue

---

### Decision · Program_Chairs · 2023-01-20

**Decision:**

Accept: poster

**Justification For Why Not Higher Score:**

The paper mainly focuses on the diversity and sensitivity of the responses, without comparison with the SotA of the open-ended dialogs, which may limit the audience.  The human evaluation is in small size and a bit weak.

**Justification For Why Not Lower Score:**

The paper is technically strong with significant improvements in terms of dialog diversity, which is an important problem in dialog research.  Four reviewers except one give quite positive recommendations.

**Metareview: Summary, Strengths And Weaknesses:**

Summary:

This paper devises a method to improve dialog generations in terms of diversity and sensitivity, by adopting a MOE structure.  Each expert is a LM which can generate utterances corresponding to a particular attribute or personality.  The expert LMs are based on a general LM with a latent variable, and conditioned on an external defined attribute or personality (sentiment, emotion, etc.).  An RL-based DM is used to select a response from the utterance generated by the expert LMs.  Evaluations are conducted with each component individually using automated metrics such as perplexity, diversity as well as human evaluation on fluency, diversity and intent precision and compared them to baselines which show promising results.

Strengths:

The paper is well motivated and clearly written.  The method is elegant and mathematically formulated.  The improvements in diversity compared to existing models are significant.

Weakness:

The main concerns raised by the reviewers include: weak baselines, the selection of automatic metrics, the human evaluations.

The authors addressed most of these concerns in the responses and revised the paper accordingly in the new version.


**Note From Pc:**

if the above contains the word "oral" or "spotlight" please see: "oral" presentation means -> notable-top-5% and "spotlight" means -> notable-top-25%. As stated in our emails, we are disassociating presentation type from AC recommendations

**Summary Of Ac-Reviewer Meeting:**

NA